# Identification of Shoot Differentiation-Related Genes in *Populus euphratica* Oliv

**DOI:** 10.3390/genes10121034

**Published:** 2019-12-11

**Authors:** Yaru Fu, Tianyu Dong, Lizhi Tan, Danni Yin, Miaomiao Zhang, Guomiao Zhao, Meixia Ye, Rongling Wu

**Affiliations:** 1Center for Computational Biology, College of Biological Sciences and Technology, Beijing Forestry University, Beijing 100083, China; fuyaruxw@126.com (Y.F.); liketianyuan@163.com (T.D.); 18851433139@163.com (L.T.); yindn_1125@163.com (D.Y.); mmzhang_2013@163.com (M.Z.); zhaoguomiao@cofco.com (G.Z.); rwu@bjfu.edu.cn (R.W.); 2Center for Statistical Genetics, The Pennsylvania State University, Hershey, PA 17033, USA

**Keywords:** shoot regeneration, differentially expressed gene (DEG), *Populus euphratica* Oliv., RNA-Seq

## Abstract

De novo shoot regeneration is one of the important manifestations of cell totipotency in organogenesis, which reflects a survival strategy organism evolved when facing natural selection. Compared with tissue regeneration, and somatic embryogenesis, de novo shoot regeneration denotes a shoot regeneration process directly from detatched or injured tissues of plant. Studies on plant shoot regeneration had identified key genes mediating shoot regeneration. However, knowledge was derived from *Arabidopsis*; the regeneration capacity is hugely distinct among species. To achieve a comprehensive understanding of the shoot regeneration mechanism from tree species, we select four genetic lines of *Populus euphratica* from a natural population to be sequenced at transcriptome level. On the basis of the large difference of differentiation capacity, between the highly differentiated (HD) and low differentiated (LD) groups, the analysis of differential expression identified 4920 differentially expressed genes (DEGs), which were revealed in five groups of expression patterns by clustering analysis. Enrichment showed crucial pathways involved in regulation of regeneration difference, including “plant hormone signal transduction”, “cell differentiation”, "cellular response to auxin stimulus", and “auxin-activated signaling pathway”. The expression of nine genes reported to be associated with shoot regeneration was validated using quantitative real-time PCR (qRT-PCR). For the specificity of regeneration mechanism with *P. euphratica*, large amount of DEGs involved in "plant-pathogen interaction", ubiquitin-26S proteosome mediated proteolysis pathway, stress-responsive DEGs, and senescence-associated DEGs were summarized to possibly account for the differentiation difference with distinct genotypes of *P. euphratica*. The result in this study helps screening of key regulators in mediating the shoot differentiation. The transcriptomic characteristic in *P. euphratica* further enhances our understanding of key processes affecting the regeneration capacity of de novo shoots among distinct species.

## 1. Introduction

A remarkable characteristic of plant cells is their totipotency, which supports the considerable regenerative capacity of plants. As an important approach of vegetative propagation in agro-forestry [1], regeneration endows plants with the ability to repair detached or wounded tissues, and also enables plants with capacity to produce new organs to achieve the opportunity of individual life. As one of the comparable ways with somatic embryogenesis in regeneration, de novo organogenesis had attracted much research focus, for example, cellular reprogramming. Extensive exploration of multilayered regulatory cascades for plant regeneration had suggested a substantial role with regeneration in plant science [2,3].

In essence, regeneration is a process in which cell fates change owing to wounding or stress [4]. Recently, massive mechanisms mediating regeneration, especially the de novo shoot regeneration of organogenesis, had been investigated. Under tissue culture condition, the transition of callus into stem cells and the subsequent differentiation of stem cells into shoot apical meristem are the two key processes during shoot regeneration [5]. For the stem cell commitment, *WUSCHEL RELATED HOMEOBOX11* (*WOX11*) and *WOX12* are involved in the first-step cell fate transition during de novo root organogenesis [6]. Zhang et al reported an essential role of *WUSCHEL* (*WUS*) in de novo establishment of shoot stem cell niche in *Arabidopsis thaliana* [7]. *CUP-SHAPED COTYLEDON* (*CUC*)*,* encoding a NAC (No Apical Meristem)-domain protein, is involved in the regulation of shoot apical meristem (SAM) formation during organ differentiation [8]. Previous research showed that *CUC* and *SHOOT MERISTEMLESS* (STM) regulate each other’s expression throughout development, but STM controlled *CUC* expression in organ boundary region [9]. As a negative regulator of meristem cell accumulation, *ULTRAPETALA1* (*ULT1*) negatively regulates *WUS* expression during meristem development [10]. To associate shoot regeneration with hormone pathways, the removal of H3K27me3 at *WUS* can be promoted by the cytokinin-rich environment, facilitating activation of *Arabidopsis* response regulators (ARRs) in cytokinin signaling pathway [7]. Members in *YUCCA* family orchestrate the endogenous auxin biosynthesis responsive for organogenesis [11]. For organogenesis of the root, gibberellin signaling is also combined with regeneration of organogenesis. Gibberellin signaling is involved in transcriptional inputs from the *SHORT-ROOT* (*SHR*), *SCARECROW* (*SCR*), and *SCARECROW-LIKE3* (*SCL3*), to regulate root formation [12]. In response to auxin, for example, lateral organ boundaries domain gene 16 (*LBD16)*, *LBD18*, and *LBD29* are important downstream genes of the auxin response factor 17/19 (ARF7/19), playing a key role in regulating lateral root formation [13,14]. 

Though the increasingly clear progress on cell fate reprograming had been achieved in studying de novo organogenesis, understanding of the regulation mechanism had still been partly summarized from model species of *Arabidopsis*. In the kingdom of plants, different species possessed highly different capacities of regeneration. For example, plants in the *Bryophyllum* family can derive a great number of shoots under a natural condition of growth. The striking difference of organogenesis capacity necessitates more insights into mechanisms of de novo organogenesis from other species in order to gain, for example, what is the key molecular processes affecting organogenesis at species level. *Populus euphratica,* an ecologically important species called “hero tree”, is widely distributed in desert area in northwest of China. Under selection of long history, *P. euphratica* had evolved an excellent system for abiotic tolerance in drought, saline, heat, and wind, making itself a model species for studying abiotic resistance mechanism [15]. Because of the harsh environment for the primary growth and propagation, studying its mechanism of de novo organogenesis should pave a way to explore how *P. euphratica* balances stress response and the preservation of regeneration capacity in further research. Besides, with the trait segregation during the sexual propagation, genotypes carrying fine trait properties can not stably pass the valuable trait to the offspring. For this reason, technology breakthrough of de novo organogenesis with *P. euphratica* should provide an effective approach for massive propagation of materials inheriting good performance in stress tolerance.

Considering the highly different organogenesis capacity among genotypes in the natural population, we select different genotypes for the shoot organogenesis test with *P. euphratica*. To get a molecular overview of organogenesis, in this study, a high differentiation group (HD) and a low differentiation group (LD) were selected. To improve our expression, we used the difference percentage of 50% at least regarding the regeneration capacity to distinguish the HD and LD. Using transcriptome sequencing and the differential expression analysis, the expression pattern between HD and LD was expected to be revealed, so as to screen genes responsible for shoot differentiation, and to further summarize the key processes affecting shoot regeneration that are specific to *P. euphratica*.

## 2. Materials and Methods

### 2.1. Selection of Genetic Lines

In 2015, natural samples of *P. euphratica* were collected from the Xinjiang Tarim River Basin, located in Luntai County, Mongolian Autonomous Prefecture of Bayingolin, 86°16’E, 41°08’N. Two hundred samples were collected with sampling sites >50 m apart. All samples were hydroponic cultivated to induce the axillary bud (3–5 cm in length) for 10 d. In plant tissue culture, after sterilization and the removal of top axillary bud, a stem segment containing two bud points was placed on the full-strength Murashige and Skoog (MS) medium (Qingdao Hope Bio-Technology Co., Ltd, Qingdao, China), supplemented with agar (Coolaber, Beijing, China) (7–7.5 g/L) + sucrose (Xilong Scientific Technology Co., Ltd, Guangzhou, China) (20 g/L) + 6-Benzylaminopurine (BBI life Sciences, Shanghai, China) (6-BA, 0.5 mg/L) + naphthaleneacetic acid (BBI Life Sciences) (NAA, 0.1 mg/L). After 20 d, newly formed buds at the shoot points of stem segments were excised (3 cm in length) and transferred to half-strength MS supplemented with agar (8 g/L) + sucrose (25 g/L) + indole-3-butyric acid (BBI Life Sciences) (IBA, 0.4 mg/L) for rooting. 

Among the 74 rooted lines, 30 were selected for adventitious bud induction experiments, with which 3–6 leaves of sterile seedlings were used as explants. To eliminate the influence of position on adventitious bud formation, the leaf tip and base (2 cm) were removed. The remaining parts were artificially damaged and transferred to MS medium supplemented with agar (7 g/L) + sucrose (25 g/L) + 6-BA (0.5 mg/L) + NAA (0.1 mg/L), with the back of each leaf facing upward, to promote differentiation of callus and adventitious buds. All samples were grown at 25 °C in the dark for 7 d, and then transferred to a 12/12 h (light/dark) photoperiod for 40 d. For each line, we observed the number of adventitious buds inducted. Using the mean shoot number and regeneration rate, HD1 and HD2 were selected to represent genotypes with high differentiation capacity, and LD1 and LD2 were selected as the low differentiation group.

### 2.2. Plant Culture and Treatment 

For each genotype, three petri dishes with each containing nine leave explants were tested for callus induction. On 25 d after the beginning of the photoperiod, when leaves differentiated into callus and adventitious buds, callus tissue excluding adventitious shoot were collected, immediately frozen in liquid nitrogen, and stored at −80°C until further use. For all four genotypes, three biological replicates were set for the further sequencing.

### 2.3. Total RNA Extraction and RNA-Seq Library Construction for Sequencing

Total RNA was extracted using the RNAprep Pure Plant Kit according to the manufacturer’s instructions (Tiangen, Beijing, China). Only RNA samples with 260/280 absorption rates of 1.9–2.0, 260/230 absorption rates greater than 2.0, and RNA integrity numbers (RINs) greater than 7 were used in subsequent experiments. Approximately 8 μg of total RNA from each sample was dissolved in Total Genomics Solution (Shenzhen, Guangdong, China). 

Total RNA was treated with deoxyribonuclease I, the bulk RNA was enriched for mRNA using oligo (dT) beads and then randomly divided into short fragments. First-strand cDNA was synthesized from mRNA using random hexamer primers. DNA polymerase I, dNTPs, and buffer were used to synthesize the second-strand cDNA. The double-stranded cDNA was purified and size-selected using the AMPure XP bead kit (Beckman Coulter, Brea, CA, USA); then, sticky ends were repaired and a single nucleotide A (adenine) addition was ligated to the sequencing adapters. The fragments obtained were selected for size and enriched through PCR amplification to create a cDNA library. An Agilent 2100 Bioanalyzer (Agilent Technologies, Santa, Clara, CA, USA) and ABI StepOnePlus Real-Time PCR System (ThermoFisher Scientific, Waltham, MA, USA) were used to evaluate the quantity and quality of RNA in each sample. The cDNA library was constructed using paired-end reads of 150 bp to obtain transcriptome data. Twelve libraries were sequenced on the Illumina HiSeq 4000 Genome Analyzer (Illumina, San Diego, CA, USA).

### 2.4. Quality Control and Calculation of Gene Expression

After filtering low quality reads, contaminants reads, and trimming adapter sequences, all subsequent analyses were based on the high-quality clean data obtained after filtering. GC content, Q20, Q30, and the sequence repetition level of the clean data were calculated. Clean reads were mapped to the *P. euphratica* reference genome (https://www.ncbi.nlm.nih.gov/search/all/?term=Populus%20euphratica) using HISAT software [16]. RNA-Seq by Expectation Maximization (RSEM) was used to estimate the expression levels using reads in the same region of different transcripts [17]. Subsequently, gene expression levels were measured and normalized using the reads per kilobase per million mapped reads (RPKM) method [18], which eliminates biased effects of sequencing depth and gene length on gene expression. The sequencing data for this study were stored in the short read archive (SRA) of the National Center for Biotechnology Information (NCBI) database, with an accession number of SRP198314.

### 2.5. Analysis of Differentially Expressed Genes 

DESeq2 package was used to analyze differential expression between HD and LD group [19]. To control false discovery rate (FDR), the Benjamini–Hochberg correction was used to adjust *p*-values of differential expression. Genes meeting a criterion of FDR ≤0.01 and fold change (FC) ≥2 were identified as differentially expressed genes (DEGs). Expression patterns of DEGs were clustered by MBCluster [20]. Gene ontology (GO) enrichment analysis of DEGs was conducted using GOseq package [21]. Metabolic pathways and signal transduction pathways in biological process-related DEGs were analyzed using Kyoto Encyclopedia of Genes and Genomes (KEGG) (http://www.genome.jp/kegg) [22].

### 2.6. Validation of Candidate Genes through Quantitative Real-Time PCR

PrimeScript RT Kit (Takara, Tokyo, Japan) was used to synthesize cDNA from 2 μg of total RNA. The Takara Ex Taq RT-PCR Kit and SYBR premix were used for qRT-PCR, which was performed on the CFX Connect Real-Time PCR System (Bio-Rad, Hercules, CA, USA). The volume for each quantitative real-time PCR (qRT-PCR) reaction was 20 μL; amplification was performed according to the following procedure: 95 °C for 30 s, followed by 40 cycles of 5 s at 95 °C and 30 s at 58 °C. A melting curve was generated at 65–95 °C to determine the specificity of each amplified fragment. For all reactions, each individual was analyzed using three technical and biological replicates. Nine candidate genes were selected for qRT-PCR analysis to validate the mRNA data. *Actin* was used as an internal control, and relative expression levels were calculated using the 2^-ΔΔCt^ method [23]. All primers used for qRT-PCR are listed in Appendix A.

## 3. Results

### 3.1. Difference of Adventitious Bud Differentiation

For the four genotypes of HD1, HD2, LD1, and LD2 with *P. euphratica,* the number of adventitious buds per explant was recorded. From the shoot number, large differentiation capabilities were observed to induce adventitious buds. HD1 and HD2 have a strong ability to form adventitious buds; their mean shoot numbers were 7.90 and 6.36, resulting in a regeneration rate of 0.96 and 0.95, respectively. LD1 and LD2 have relatively low differentiation ability. Starting from an equal number of explant number, LD1 and LD2 showed mean shoot numbers of 0.28 and 0.53, respectively, the regeneration rate of 0.21 and 0.33, respectively, were found (Figure 1). The difference analysis on the regeneration rate indicated that the genotype highly affected adventitious bud formation.

With three biological replicates settled for each genotype, 12 libraries of *P. euphratica* were sequenced at the transcriptome level. The correlation coefficients among the expression data provided by the four genotypes showed that HD1 and HD2 clustered together, and LD1 and LD2 were clustered into a second group. Each of the two clusters had a high within-cluster correlation of samples, above 0.97 and 0.92, respectively, but the correlation between cluster 1 and cluster 2 was weak, at below 0.88 (Figure 2). The pairwise expression correlation with distinct samples had confirmed an establishment of the differentiation group, that is, high differentiation including HD1 and HD2, and low differentiation including LD1 and LD2. It is suggested that transcriptome sequencing using these two hugely differentiated groups of genotype can provide a direct insight into the de novo organogenesis of the shoot. 

### 3.2. Differentially Expressed Genes

Comparing the expression between the HD and LD group, using a criterion of FDR ≤0.01 and FC ≥2; a total of 4920 genes were identified as DEGs. Of them, 1654 were up-regulated in the HD group, while 3266 were down-regulated. The significant differences in gene expression between HD and LD suggested that DEGs might be involved in the molecular regulation of shoot differentiation. Using these DEGs, the clustering algorithm revealed five classes of the expression pattern in total. On the whole, the logarithm of FC with expression relative to the mean expression provides three clusters of down-regulation and two clusters of up-regulation.

Cluster 2 and 5 groups in Figure 3B,E are the strikingly changed expression pattern, each containing 570 and 629 genes. For cluster 2, which is down-regulated in the LD group, 55 transcription factors (TFs) proteins were revealed, comprising 15 *AP2/ERF* family proteins, 13 members within the *bHLH* family, *7 MYB* family genes, and 20 other TFs belonging to 13 TF family. For cluster 5, which is highly expressed in the LD group, 8, 7, 7, 4, and 3 genes in *LOB, MYB*, *NAC*, *ERF, WRKY* family, respectively, and other important TFs comprised 1 WUSa, 1 NGA4, and 1 *JUNGBRUNNEN 1-like*. Among genes in cluster 2, 36 were involved in plant hormone signal transduction, of which 7 were covered in auxin transportation, auxin-responsive, and auxin-induced protein. Besides the auxin- and ethylene-related genes in cluster 5, a pyrabactin resistance protein inhibiting growth was found. Five senescence-associated genes were also detected to be up-regulated in LD (Figure 3 and Appendix A).

### 3.3. Functional Classification of DEGs using GO and KEGG Pathway Analysis

To further understand functions with DEGs, GO analysis was carried out to explore the core classification of regulation. For biological processes, the enrichment using all 4920 DEGs provided 15 significant terms (*p* < 0.01), including “cell differentiation”, “regulation of hormone levels”, “auxin-activated signaling pathway", “cellular response to auxin stimulus", and “developmental process" (Figure 4 and Appendix A). The results showed a vital role of auxin and the abundance accumulation of hormone in regulating de novo organogenesis. Specifically, the GO term of “cell differentiation” included 261 genes, from which some differentiation genes for determining cell fate can be screened, such as *brassinosteroid insensitive 1*, *YABBY*, and *ARF5*. For cellular components, auxin-related genes, that is, genes encoding the auxin efflux and influx carrier protein, were involved in the plasma membrane part, also suggesting an indispensable regulation of auxin for regeneration of the adventitious bud.

Similar to the results of GO enrichment, enrichment analysis of the KEGG pathway with DEGs equally provided significant pathways of “plant hormone signal transduction”. Among the highly enriched pathways, except biosynthesis pathways and metabolism pathways, “mitogen-activated protein kinase (MAPK) signaling pathway–plant” is the top noteworthy pathway, because genes within this pathway participated in signal transduction related to cell proliferation, cell growth, and cell differentiation. DEGs for the MAPK signaling pathway in this study involved heat shock protein 70 (HSP70), serine/threonine/tyrosine kinase family protein gene, mitogen-activated protein kinase kinase kinase (MKKK), mitogen-activated protein kinase kinase (MKK), and mitogen-activated protein kinase (MAPK). “ABC transporters” and “brassinosteroid biosynthesis” are the other two significantly enriched pathways, indicating a vital role of the hormone in mediating shoot regeneration (Appendix A and Appendix A).

### 3.4. Shoot Differentiation Related Genes

In order to know whether DEGs participated in the regulation of organgenesis, by summarizing reported genes in recent studies, we identified their homologous genes that might also be involved in the do novo shoot organogenesis in *P. euphratica*. To sum up, different classes are categorized as auxin pathway genes, cytokinin pathway genes, SAM related genes, cell cycle control genes, and senescence genes. For DEGs in cluster 2 and cluster 5, different classes of regeneration-related genes are given in Table 1. Senescence genes in cluster 5 showed an average of 2.22 up-FC in the LD group.

Among 40 differentiation-associated genes, auxin-related genes comprising genes of auxin response, auxin transport, auxin stimulate response were significantly considered to contribute their key roles in shoot formation. For transport of auxin, proteins in PIN1 are important for the auxin distribution and patterning in cluster 1 and cluster 3. *YUCCA* family genes that responsible for auxin production were revealed in cluster 1. For cytokinin related genes, *ARRs* are regulators of cytokinin signaling. Both *ARR5* and *ARR8* were down-regulated in cluster 3 to control the shoots’ differentiation. Four *LOG* genes participating in cytokinin signals were found to be down-regulated in HD genotypes. To associate with gibberellin signaling, transcriptional inputs from the *SCARECROW* (*SCR*) and *SCARECROW-LIKE3* (*SCL3*) genes in GRAS subfamilies were found in clusters 1, 2, and 3. For cell fate determination, *CLV*, WOX, and *WUS* genes were found in cluster 1, 2, and 3. Diagram integrating roles of the auxin signaling pathway, cytokinin signaling pathway, and wound response are shown in Figure 5. Specially, the wound-induced genes, like ERF109, showed a nearly −6.5 FC of upregulation in the HD group. For genes functioning in the meristem, *AINTEGUMENTA* (*ANT*) genes in the (APETALA2/Ethylene Response Factor) AP2/ERF family were discovered to be down-regulated in cluster 1. *CUC* genes discovered in the *NAC* family within DEG cluster 2 control shoot organ boundary formation, which showed a −6.93 FC. To control the cell cycle, KNOX protein from cluster 1 and 2 was reported to act within the meristem to promote cell division, and *CDKs* were revealed to show a down-regulated expression in the HD genotype.

### 3.5. Other Potential Key Genes

To reveal some transcriptional features specific to *P. euphratica*, besides the above listed differentiation related genes, we noticed 88 DEGs participating in ubiquitin-26S proteosome mediated proteolysis pathway that distributed in all five clusters, for example, U-box or F-box domain containing protein, SKP1, E3 ubiquitin ligase of BIG BROTHER-like protein, PUB14, LIN, RGLG2, FANCL, SINAT2, MARCH1, and ATL6. Most of them mediated proteolysis through ubiquitination of the target protein, which was further recognized and degraded by 26S proteasome. The large number of genes belonging to this pathway indicated a common pathway of protein degradation in regulating organogenesis. From the significantly enriched GO terms, we also marked the “plant–pathogen interaction”, which involved 314 key genes with expression changed significantly, encoding WRKY, LRR receptor protein, disease resistance protein, TIR-NBS disease resistance, and NBS-LRR type disease resistance protein, respectively. Besides the auxin and cytokinin related genes, brassinosteroid responsive, abscisic acid, and insensitive genes were both identified as down-regulated expression genes in the LD group in cluster 1, cluster 2, and cluster 3. Gibberelin receptor GID1, gibberellin 20-oxidase gene, and gibberellin regulated genes were found in cluster 4 to show an at least two-FC of the expression in LD, but chitin-inducible gibberellin-responsive gene in cluster 1 had a five-FC of down-regulation.

Interestingly, stress-responsive DEGs encoding universal stress protein, dehydration responsive element binding family protein (DREB), CBL-interacting protein, heat stress TF family protein and stress responsive A/B domain family protein, 33 ethylene responsive element binding factor family genes, and 3 ethylene responsive factors were found. Accompanied by the discovery of stress responsive genes, we screened 17 genes associated with senescence. Two of them in cluster 2 showed a very low expression abundance. Cluster 1 and cluster 3 included two and four senescence-associated genes, showing a 6.5–7.4-fold and 2.27–3.7-FC, respectively, of down-expression in the LD group. For the up-regulation of expression in the LD group, four and five genes were revealed to have a 1.9–2.8-fold and 3–6.46-FC in cluster 4 and cluster 5, respectively. The disclosure of genes in this class might indicate an indirect relationship for balancing senescence and differentiation.

### 3.6. Quantitative Real-Time PCR Validation of Shoot Differentiation-Related Genes

From the differential expression analysis, 11 DEGs closely related to the differentiation response were selected to validate the expression accuracy detected by RNA-seq (Figure 6A–K). Primer sequence information obtained are given in the Appendix A (see Appendix A). In all four genotypes, LOC105115136 (ANT), LOC105112164 (CUC), LOC105121805 (SCR1), LOC105142364 (SCR2), LOC105108097 (ARR), LOC105140873 (LBD), LOC105142556 (LOG), and LOC105128586 (DREB) exhibited significant differences between the HD and LD group. Their expression in HD was much higher than that in LD. In contrast, LOC105139456 (a cyclin-dependent kinase, CDK), LOC105136218 (ULT1), and LOC105142505 (WIND) showed an opposite expression pattern. The correlation coefficient of the expression levels between qRT-PCR and RNA-seq was 0.8596 (p ≤ 0.01), suggesting reliability of the RNA-seq data (Figure 6L).

## 4. Discussion

The initiation of organ growth and development of plants throughout life cycle is closely related to activities in apical meristem [24]. Shoot apical meristems (SAMs) harbor a small set of stem cells located at the tip of each plant, which integrated a complex, dynamic, and interconnected network of different cell types to regulate plant growth and development [25]. Whether leaves differentiate into callus, and when to activate differentiation, are regulated through a series of genetic and environmental interaction. The underlying and interacting molecular mechanisms of shoot differentiation play crucial roles in plants, and have currently become an important and popular focus of research into plant regeneration and genetic transformation. Therefore, identifying genes involved in the regulation of shoot regeneration is essential for understanding the intrinsic mechanism driving these processes. 

Using transcriptome sequencing of four genotypes that hugely differed in differentiation capacity, the study provides a comprehensive understanding into molecular mechanism of de novo organogenesis in *P. euphratica*. During the de novo organogenesis process, explants of leaves differentiate into callus first, and then form adventitious buds [11]. On the basis of DEGs, the clustering algorithm revealed a total of five expression patterns between HD and LD genotypes. Using the classic method of GO enrichment and KEGG enrichment, auxin-related terms including “auxin-activated signaling pathway”, “cellular response to auxin stimulus”, and “plant hormone signal transduction” showed a crucial role of auxin and its abundance in regulating de novo organogenesis of shoot. Together with the significant term of “cell differentiation”, the alignment of genes with other research reports identified seven classes of differentiation-associated genes. Each of them coordinated to play various roles in shoot regeneration. Furthermore, we also identified some specific classes of genes for *P. euphratica*. Using materials sampled from natural population, the inherent difference of differentiation capacity with different genotypes in this study provided an overall landscape of organogenesis mechanism at the transcriptome level.

Compared with the reported identification of genes in differentiation regulation, the seven classes of key genes were first screened out to explain a similar regulation mechanism of shoot organogenesis in *P. euphratica*, although different genotypes in natural population had huge differences in genetic backgrounds. In detail, auxin-related genes including auxin responsive genes and auxin regulated factors are important for the formation of lateral organs [13,14]. Auxin carrier genes, that is, *PINs*, are important for the auxin distribution and patterning of the de novo meristem [26]. Expression of SAM related genes and *NAC* genes support their previously reported role in determining the cell fate alteration and organ boundary formation through development [27,28,29]. *Gong* et al. found role of *SCR* and *SCL3* in gibberellin for regulating root formation, but the identification of them as DEGs in this study also indicates their significant role in shoot formation [12]. The identification of cytokinin pathway genes, *GRAS* family gene, *AP2/ERF* family gene, *KNOX*, *CDK*, *YABBY*, and *YUCCA* as DEGs, and the further expression validation of them using qRT-PCR, are likely to promise their consistent roles in shoot regeneration with *P. euphratica*. Taking CDK as an example, which is essential to regulate G1/S and G2/M transformation in cell division during all stages of cell cycle, is equally indispensable for cell expansion in leaf, plant cell signal transduction, and regulation of differentiation [30]. In this study, the up-regulated expression of *CDK* in LD group potentially indicates that *CDK* overexpression causes plants lose the ability to form callus or regenerate shoots. One possible reason is that the disruption of normal *CDK* expression might render cells unable to properly respond to hormonal stimulation during organogenesis [31].

On the basis of existing expression difference between HD and LD, some transcriptome feature that are specific to *P. euphratica* were summarized. First, the large number of DEGs revealed in ubiquitin-26S proteosome mediated proteolysis. Within this pathway, genes encoding extensive E3 ubiquitin protein ligase and proteins constituting the different types of SCF complex and the U-box type E3 were identified differently expressed. Although proteins targeted to be ubiquitinated are not clear during organogenesis, through this pathway, the corresponding recognition pair between the target protein and distinct type of E3 still needs to be clarified. The pathway may aid the perception of key signals through rapid degradation after polyubiquitination to regulate downstream pathways [32], as auxin signalling pathway and the cytokinin perception pathway had essential regulatory functions on shoot regeneration. Studies evidenced AUX/IAA may dissociate ARF, which regulates the regeneration capacity significantly [33,34]. Other studies also support the ubiquitination of AUX/IAA, with which the activity of ARF might be regulated. Still, in the cytokinin signalling pathway, we found E3ligase of MMS21, and PUB2/4 [35,36], both of which regulate activity of ARR, which promotes the shoot regeneration.

Second, for the significantly enriched GO term of “plant–pathogen interaction”, a large number of disease resistance genes were screened out. However, by referring to the literature, a direct relationship between shoot organogenesis and disease resistance cannot be established. Yet, plants launch organogenesis programme when they suffer wounds. This transcriptomic feature might be explained as an accompanying effect of plant immune system activation [37]. Plants regularly endure wounds caused by abiotic or biotic environmental stimuli and have developed extraordinary abilities to restore their tissues after injuries [38]. In cluster 4, a wound-responsive family protein that causes plants to form a defensive and protective mechanism was discovered. The wound on leaf explant potentially induces plants' defense and immune response, which indicated a non-nuclear role of disease resistance for organogenesis procedure, but a crucial role for the secure maintenance of callus tissue for the transition into stem cells. Plants are able to sense the injured tissue as an altered self and induce a response similar to those activated by pathogen infection. Wounded tissue may release molecules as microbe-associated molecular patterns (MAMPs) or damage-associated molecular patterns (DAMPs) that activate the plant innate immunity [39]. The induced immune response functions to sterilize the site of injury and remove dead or dying cells, securing the maintenance of normal callus tissue, which constitutes a premise step for the cell fate conversion of stem cells. As a common role of jasmonate (JA) acting in pathophysiology and wound signaling [40], JA promotes stem cell activation and regeneration in stem cell niche through ERF115 [41]. In our study, two genes of ERF109 showed a −6.43 and −6.50 FC of down-regulation in the LD group. Zhang et al. also uncovered the activation of ERF109 by JA, which further upregulates the ANTHRANILATE SYNTHASE α1 (ASA1)-a tryptophan biosynthesis gene in the auxin production pathway, promoting the de novo regeneration of root [42]. Therefore, the coordination between wound response or wound-induced immunity and regeneration might be built through JA signaling. In addition, the DEG result containing stress-responsive genes suggests an indirect relationship between stress response and shoot regeneration. DREB2A, so far reported its role in tolerating drought and heat stress, but, using an enhanced yeast one-hybrid screen method, Ikeuchi et al. revealed that *DREB2A* can bind to the promoters of *WOUND INDUCED DEDIFFERENTIATION 2* (*WIND2*) and *WIND3*, suggesting that *DREB2A* is also involved in wound induced cellular reprogramming [43].

Third, in this study, DEGs related to senescence were identified, suggesting that cell senescence might also be involved in meristem differentiation. Senescence is a final stage of growth or development, but is often induced prematurely following exposure to stresses [44]. Therefore, from the results of the significant occurrence of the “plant–pathogen interaction”, DEGs of stress-associated genes and DEGs relating to senescence should be systematically interconnected. The callus-inducing medium (CIM)-induced organogenesis possibly had general similarity with wound-induced organogenesis [45]. As Seen from the results in this study, we know the molecular mechanism between both manners of shoot regeneration all involved in stress-responsive programme. Senescence had a close regulatory roles regeneration [46]. Rapp et al. reviewed many attributes common to senescence and dedifferentiation, so the differentiation ability was hugely affected by the senescence programme [44]. In the LD group, the high expression with some of the senescence-associated family gene might be caused by the stress-responsive mechanism that induced by the CIM manner of organogenesis. Such activated expression of senescence-related genes can possibly account for the low level of shoot regeneration in LD genotypes, which might answer to the critical question of key processes that affecting difference in regeneration capacity.

## 5. Conclusions

To understand the molecular mechanism of shoot organogenesis with *P. euphratica*, from the nature population, we selected genotypes with distinct differentiation capacity to be sequenced at the transcriptome level. During the shoot regeneration process, DEGs involved in “cell differentiation”, “regulation of hormone levels”, “auxin-activated signaling pathway”, “cellular response to auxin stimulus”, and “plant hormone signal transduction” were enriched. By comparing expression data between HD and LD, several classes of differentiation-related genes were identified, the expressions of which were further validated by qRT-PCR, suggesting a similar mechanism of organogenesis in *P. euphratica*. For shoot organogenesis, the transcriptome investigation additionally provided specificity of shoot regeneration with *P. euphratica*, for example, the large number of DEGs in ubiquitin-26S proteosome mediated proteolysis. The results with the top enriched term of the “plant–pathogen interaction”, and identification of wound-responsive DEGs, stress-responsive DEGs, and senescence-associated DEGs possibly accounted for the differentiation between distinct genotypes. The results provided insight into the molecular mechanism potentially controlling shoot differentiation. Undoubtedly, the future dissection into the functions and potential interactions between differentiation genes elucidated at the molecular level should further improve understanding of this process.

## Figures and Tables

**Figure 1 genes-10-01034-f001:**
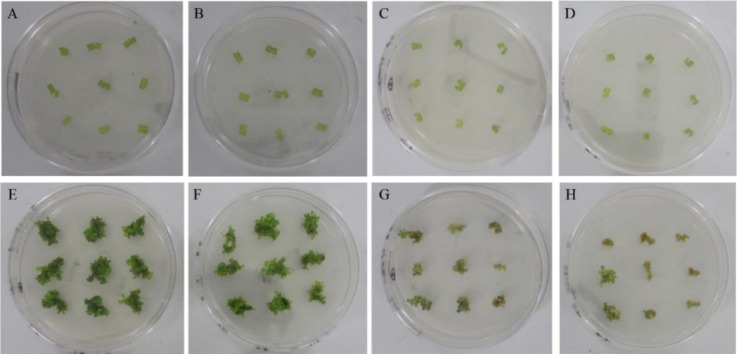
Shoot regeneration with four genotypes of *Populus euphratica*. (**A**–**D**) are leaves’ explants inoculated on media for each of the highly differentiated group 1 (HD1), HD2, low differentiated group 1 (LD1), LD2, respectively. (**E**–**H**) indicate regeneration of adventitious buds at day 32 of the HD1, HD2, LD1, LD2, respectively.

**Figure 2 genes-10-01034-f002:**
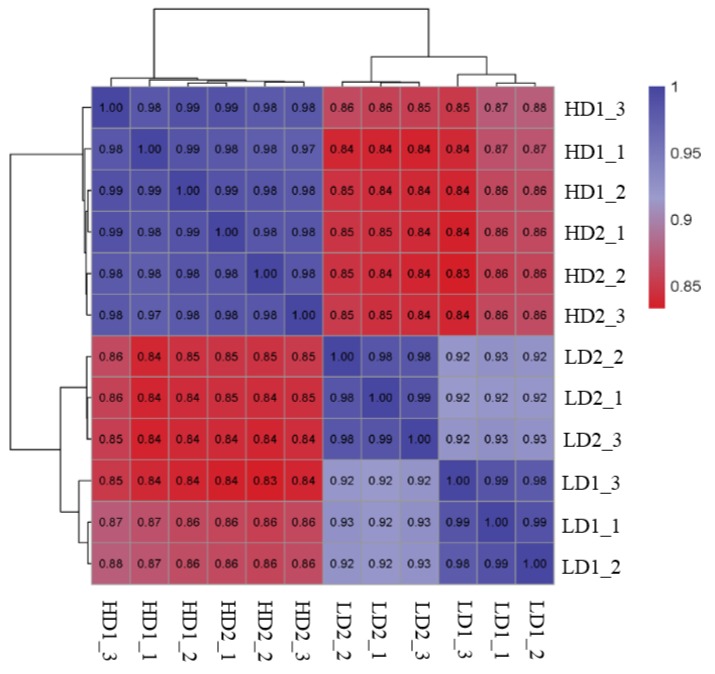
The pair-wise correlation of transcriptome expression among 12 libraries of *P. euphratica*, including four genotypes, each containing three replicates.

**Figure 3 genes-10-01034-f003:**
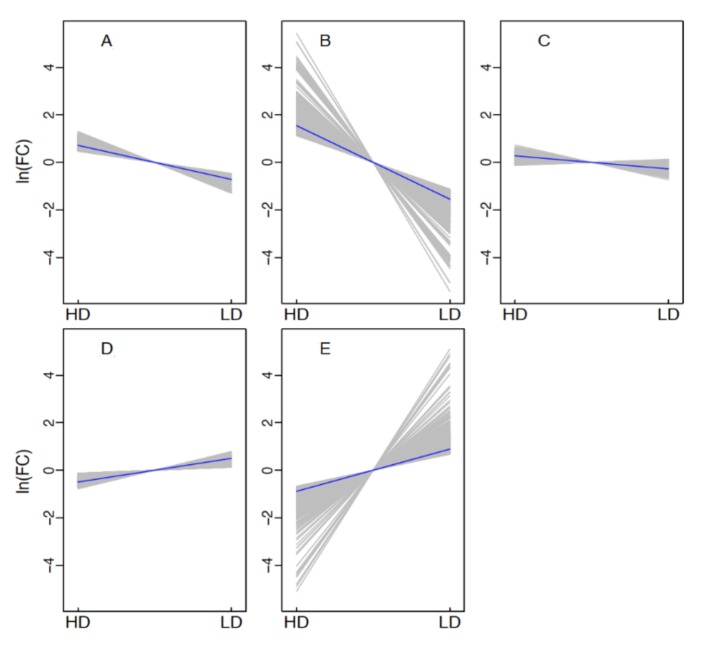
Cluster analysis of the differentially expressed genes’ (DEGs’) expression patterns in both HD and LD. The Y axis represents logarithm of fold change (FC) with expression levels relative to the mean of the same gene. (**A**–**E**) indicate clusters 1–5, respectively.

**Figure 4 genes-10-01034-f004:**
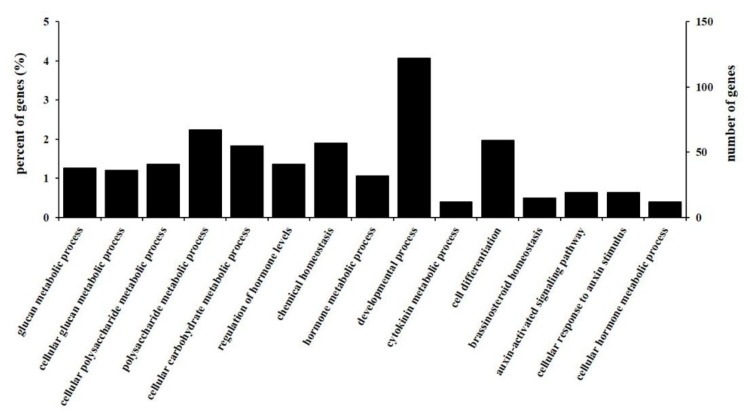
Biological process (BP) of Gene Ontology (GO) classification of differentially expressed genes (DEGs) between HD and LD. The right Y-axis represents the number of DEGs in a category; the left Y-axis indicates the percentage of a specific category of DEGs in each main category.

**Figure 5 genes-10-01034-f005:**
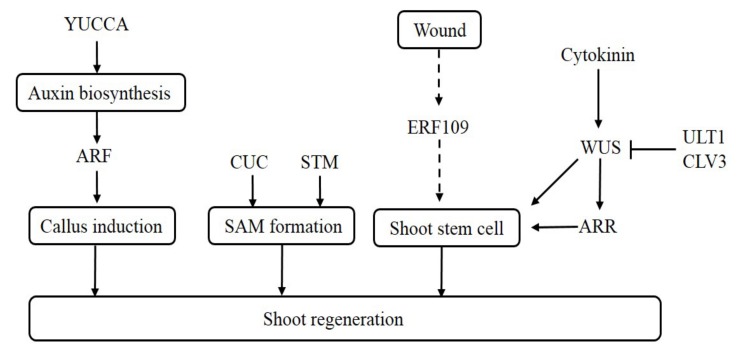
Pathways mediating shoot regeneration with *P. euphratica*. STM, shoot meristemless; SAM, shoot apical meristem; ARF, auxin response factor; WUS, wuschel; ULT, ultrapetala; ARR, *Arabidopsis* response regulator.

**Figure 6 genes-10-01034-f006:**
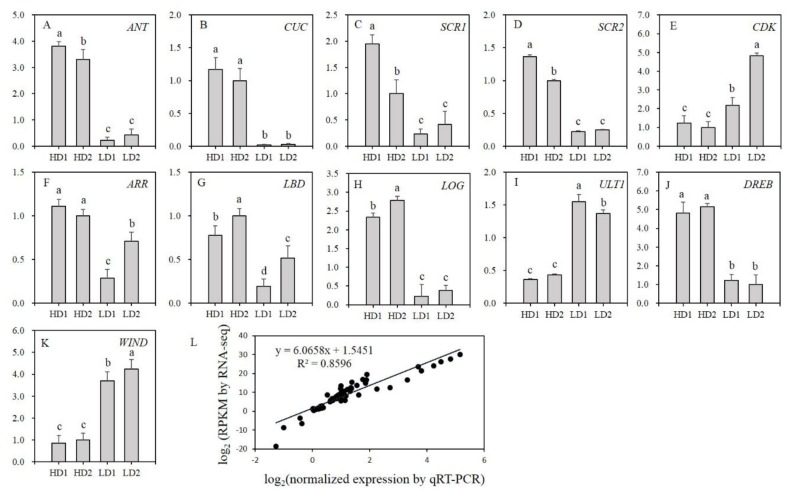
Quantitative real-time PCR (qRT-PCR) validation of 11 gene expression that are involved in shoot differentiation. (**A**–**K**) indicate the normalized expression pattern of the 11 DEGs. (**L**) indicates correlation of the expression obtained from qRT-PCR and RNA-Seq. Note: the normalized expression with the Y-axis was log_2_ transformed. Different letters in superscript indicate a significant difference between any pair of genotypes at a significance level of 0.01. DREB, dehydration responsive element binding family protein; LBD, lateral organ boundaries domain; CDK, cyclin-dependent kinase; SCR, scarecrow; ANT, aintegumenta.

**Table 1 genes-10-01034-t001:** Homologous genes involved in shoot regeneration with *Populus euphratica*. SAM, shoot apical meristem.

Gene classes	Gene Name	Cluster	log2FC
**Auxin related genes**	*Aux28* *ARF* *IAA29* *LBD12* *LBD16* *LBD41*	Cluster 2 Cluster 2 Cluster 2 Cluster 5 Cluster 5 Cluster 5	−8.27 −4.53 −4.21 1.95 3.74 3.28
**SAM related genes**	*ANT* *WOX1a* *CUC* *SCR* *WUSa*	Cluster 2 Cluster 2 Cluster 2 Cluster 2 Cluster 5	−4.57 −6.40 −6.93 −5.96 2.45
**Cell cycle control genes**	*KNOX*	Cluster 2	–5.81
**Wound- or stress-induced genes**	*ERF109* *WRKY41* *WRKY53* *DREB* *WRKY22* *WRKY47*	Cluster 2 Cluster 2 Cluster 2 Cluster 2 Cluster 5 Cluster 5	−6.50 −4.88 −3.90 −6.80 1.61 2.17
**Senescence genes**	*SENESCENCE101* *SENESCENCE20* *SENESCENCE1*	Cluster 5 Cluster 5 Cluster 5	2.21 2.78 1.68

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
