# Peer review of "Identification of Shoot Differentiation-Related Genes in Populus euphratica Oliv."

_genes, 2019, doi:10.3390/genes10121034_

Round 1
Reviewer 1 Report
This manuscript describes a study that identified genes involved in the process of shoot differentiation during genetic transformation of Populus euphratica. The study focuses on shoot regeneration during genetic transformation, which is important, but fails to address de-novo shoot regeneration as part of asexual reproduction as mentioned in the very first sentence of the abstract and which the introduction seems to indicate would be the focus of the paper. Given the body of the manuscript the abstract and introduction should be re-worked to better reflect the study.
Significant Changes:
It’s unclear from the manuscript whether genotypes were designated either HD or LD or whether individuals from each genotype were assigned to each designation. Further, the HY1-4 designations are also confusing. Are these a refinement of the HD LD designations or do these refer to something else entirely.
Line 198-208: When talking about the transcription factors (TFs), listing the transcription factor families (TFFs) is not helpful. The number of TFs associated with each TFF should be indicated. Identifying a single TF in a given TFF might not be interesting, but identifying 15 TFs in a single TFF might be.
Section 3.3 (starts with line 212): This is a literary review of the known roles of auxin in de-novo differentiation. While interesting, it provides no new findings or insights learned from the experiment the manuscript is based on. Given that a number of the genes discussed in this section were found to be differentially expressed the authors could propose and illustrate a pathway that illustrates how the differentially expressed genes are all involved/interwoven.
Line 243: Again, it’s common knowledge that cytokinin is important in plant transformation. What the data generated from this experiment could add to the body of knowledge is the gene expression associated with the process (ie: draw a pathway and denote each step as up or down regulated based on transcriptome data). As presented the data is all there, but it’s not easy to interpret the biological importance of the findings.
Section 3.6 (starts line 291): This section seems to be cherry picking genes of interest. These genes are already known to be involved in differentiation, so while I appreciate their inclusion, but I would have liked novel genes associated with differentiation included as well.
Starting line 352: Ubiquitination targets proteins for degradation. It’s unclear, based on the manuscript, how this pathway is associated with de-novo organogenesis.
Line 366-369: The leaf explant is wounded, therefore it mounts a wound response. That is what I would expect. I fail to see the link to non-nuclear role of disease resistance for organogenesis and a role to the secure maintenance of callus tissue into stem cells. This seems pure conjecture and is not backed up by any experimental evidence.
Line 377: I don’t think the citation is correct.
Line 382-383: I don’t believe senescence is a process of trans-differentiation a transition from one cell type to another. I looked at the reference (45) and didn’t think that’s what the reference said either. Please fix or better address.
Figure 4. The figure legend is not informative at all. Only the lowest adjusted P-value (value not provided) seems as though it would be significant. Differentially expressed genes can be assigned GO association in both biological process AND molecular function. Therefore, the authors should pick one of the classifications, not all three (cellular component is usually the least informative category).
Table 1: This is not an informative table. Genes associated with each class are distributed across both HD and LD genotypes. What’s interesting is that only YUCCA genes are unique to cluster 1 (HD).
Minor Changes:
Lines 86-87: There should be numbers associated with the differentiation capacity of ‘highly differentiated’ and ‘lowly differentiated’ groups.
Line 45: ‘regeneration is a process in which cell fates change due to wounding or stress’ That is not what reference 4 says, in fact that is a pretty broad/obscure definition of regeneration.
Line 145: The SRA accession number ‘can’t be found’. Please double check the number, but it might also be more applicable to provide the BioProject ID.
Figure 3 the labels are 1-5, but in the text they are referred to as A-E.
Finally, there is no confirmation that the DEGs discussed in this manuscript are involved in the shoot differentiation process. Further, there is no compelling story that leads me to believe that these genes are involved in the differentiation process.
Author Response
Reviewer #1
This manuscript describes a study that identified genes involved in the process of shoot differentiation during genetic transformation of P. euphratica. The study focuses on shoot regeneration during genetic transformation, which is important, but fails to address de-novo shoot regeneration as part of asexual reproduction as mentioned in the very first sentence of the abstract and which the introduction seems to indicate would be the focus of the paper. Given the body of the manuscript the abstract and introduction should be re-worked to better reflect the study.
Our response: Thank this reviewer for his/her enormous time and effort given to review our manuscript. S/he provided an unusually massive amount of reviews which we feel very useful for revising our manuscript and improving its presentation.
Again, we deeply appreciate the positive view from reviewer#1 that shoot differentiation had valuable applications in both genetic transformation and asexual reproduction. For the study in this manuscript, shoot regeneration-related genes were identified by analyzing transcriptome from high differentiation (HD) and low differentiation (LD) group. In essence, what we are focusing on is the transcriptome pattern affecting capacity of de-novo shoot regeneration, but not for the experiment of genetic transformation. We're afraid s/he might had misunderstood our intended meaning, which may largely be caused by our inappropriate word expression. To avoid the ambiguity of expression, for the first sentence in abstract, we had rewrite it according to its importance in plant biology, see line 13-16. And in introduction part, we double checked there is no content regarding the shoot regeneration for genetic transformation or asexual reproduction.
It’s unclear from the manuscript whether genotypes were designated either HD or LD or whether individuals from each genotype were assigned to each designation. Further, the HY1-4 designations are also confusing. Are these a refinement of the HD LD designations or do these refer to something else entirely.
Our response: By using HY1-4, readers cannot designate each genotype to either HD or LD, we had now realized this problem of sample name. To avoid the confusing situation, sample name of HY1-4 were already changed as HD1, HD2, LD1, LD2 respectively. HD1-2 represent highly differentiated genotypes, LD1-2 represent lowly differentiated genotypes. All expression related the sample name, or that occurred on figures had all been changed, for example, line 104-105, line 158-162, line 167-175. Also labels for sample name in figure 2 had been changed, see line 178-179.
Line 198-208: When talking about the transcription factors (TFs), listing the transcription factor families (TFFs) is not helpful. The number of TFs associated with each TFF should be indicated. Identifying a single TF in a given TFF might not be interesting, but identifying 15 TFs in a single TFF might be.
Our response: We follow the reviewer#1's advice. Not only TFF name, but also TFFs only containing large number of TFs were screened out to highlight their potential role for regeneration. The number of TFs associated within each TFF screened newly was indicated on line 191-195.
Section 3.3 (starts with line 212): This is a literary review of the known roles of auxin in de-novo differentiation. While interesting, it provides no new findings or insights learned from the experiment the manuscript is based on. Given that a number of the genes discussed in this section were found to be differentially expressed the authors could propose and illustrate a pathway that illustrates how the differentially expressed genes are all involved/interwoven.
Line 243: Again, it’s common knowledge that cytokinin is important in plant transformation. What the data generated from this experiment could add to the body of knowledge is the gene expression associated with the process (ie: draw a pathway and denote each step as up or down regulated based on transcriptome data). As presented the data is all there, but it’s not easy to interpret the biological importance of the findings.
Our response: Thanks for this constructive suggestion for using a pathway diagram to indicate genes involved in auxin and cytokinin pathway. We combined both auxin and cytokinin pathways into one and the same picture, which was arranged in figure 5. For new findings, we had implemented the connection between wound signaling and activation of shoot stem cell. Thus, auxin pathway, cytokinin pathway and wound signaling were integrated to account for the molecular mechanism of shoot regeneration together. Corresponding description with section 3.4 had been changed, see line 236-254.
Section 3.6 (starts line 291): This section seems to be cherry picking genes of interest. These genes are already known to be involved in differentiation, so while I appreciate their inclusion, but I would have liked novel genes associated with differentiation included as well.
Our response: In section 3.6 of "qRT-PCR of shoot differentiation-related genes", for novel genes that might be involved in shoot regeneration with our experiment on P. euphratica, as discussed on line 371-373 with our original manuscript, DREB, WIND had potential to correlate their function with shoot regeneration. To further illustrate their biological role, we implemented their expression validation. You will see the change with Figure 5. Yet, because of the new complement of a new diagram indicating the auxin and cytokinin pathway impacted on shoot regeneration, figure 5 had been renamed as figure 6. Following the change in figure, we had made some word modification in section 3.6, as marked in red with line 281-294.
Starting line 352: Ubiquitination targets proteins for degradation. It’s unclear, based on the manuscript, how this pathway is associated with de-novo organogenesis.
Our response: For de-novo organogenesis of shoot, as summarized in result section with the manuscript, auxin signalling pathway as well as the cytokinin perception pathway had essential regulatory functions on shoot regeneration. Many studies evidenced AUX/IAA may dissociate ARF, which regulate the regeneration capacity significantly. Literatures also support the ubiquitination of AUX/IAA, with which, the activity of ARF might be regulated. Still, in cytokinin signalling pathway, we found E3 ligase of MMS21, PUB2/4, both of them regulate activity of ARR, which promotes the shoot regeneration. To discuss how ubiquitination for protein degradation associate with the de-novo organogenesis, we had complemented these information in manuscript, seen on line 343-347.
Line 366-369: The leaf explant is wounded, therefore it mounts a wound response. That is what I would expect. I fail to see the link to non-nuclear role of disease resistance for organogenesis and a role to the secure maintenance of callus tissue into stem cells. This seems pure conjecture and is not backed up by any experimental evidence.
Our response: For the link between disease resistance for shoot regeneration and maintenance of callus tissue, actually, we didn't find some direct backup indeed. To improve our expression accurately, we had added sentences to explain the similarity between wound-induced response and the pathogen infection-induced response, which constitutes a premise for the normal induction of callus. Also role of Jasmonate on stem cell activation and regeneration had been discussed, as seen on line 357-368.
Line 377: I don’t think the citation is correct.
Our response: To support statement of senescence can often be induced prematurely following exposure to stresses, after carefully checking the reference list, we found we had mistakenly arranged their order. We originally intended to cite the 46-th reference of "Senescence Meets Differentiation" by Rapp Y et al. Now we had corrected its order as 43.
Line 382-383: I don’t believe senescence is a process of trans-differentiation a transition from one cell type to another. I looked at the reference (45) and didn’t think that’s what the reference said either. Please fix or better address.
Our response: Yes, we apologize for our deficient expression of senescence. Actually, we were aiming to convey the opinion that senescence had a close regulatory roles regeneration, as evidenced by a new reference, "Cellular senescence in development, regeneration and disease" by Muriel Rhinn et al. The corrected expression can be seen as line 381. The corrected reference also can be found on line 504-505.
Figure 4. The figure legend is not informative at all. Only the lowest adjusted P-value (value not provided) seems as though it would be significant. Differentially expressed genes can be assigned GO association in both biological process AND molecular function. Therefore, the authors should pick one of the classifications, not all three (cellular component is usually the least informative category).
Our response: To make figure 4 more informative, we were listed a little of MF and CC classification of GO, to highlight the roles provided by BP terms. More BP terms with distinct P-values were additionally presented with figure 4. Because of the removal with MF and CC, we have newly changed its format as a bar plot, as seen on line 214-218.
Table 1: This is not an informative table. Genes associated with each class are distributed across both HD and LD genotypes. What’s interesting is that only YUCCA genes are unique to cluster 1 (HD).
Our response: Thanks for the suggestion. The purpose of Table 1 is to list important genes related to bud differentiation from five clusters. But genes unique to any one of the clusters should provide the most informative knowledge to readers. Since different members within the same family had distinct biological functions, we had carefully re-screened organogenesis-related genes, from which unique expression patterns were newly determined in Table 1. The corrected can be seen as line 231-251.
Lines 86-87: There should be numbers associated with the differentiation capacity of ‘highly differentiated’ and ‘lowly differentiated’ groups.
Our response: Thanks this reviewer for his/her careful review. Because this query was raised in Introduction part, readers commonly do not have a perception to what a degree materials can be designed into 'highly differentiated (HD)' group, and what degree materials can be grouped into 'lowly differentiated (LD)' group. To improve our expression, we use the difference percentage of 50% at least regarding the regeneration capacity to distinguish the HD and LD. We had changed the sentence to 'a high differentiation group (HD) and a low differentiation group (LD) were selected, with their regeneration rate diverged 50% at least', as shown on line 82-83.
Line 45: ‘regeneration is a process in which cell fates change due to wounding or stress’ that is not what reference 4 says, in fact that is a pretty broad/obscure definition of regeneration.
Our response: For the broad definition of regeneration, we have cited a new reference "Stem cells and plant regeneration" published on Development Biology in 2018. Please refer to the reference section in revised manuscript on line 418 for the new reference [4].
Line 145: The SRA accession number ‘can’t be found’. Please double check the number, but it might also be more applicable to provide the Bio Project ID.
Our response: During the review stage, data of clean sequence were transferred to the SRA deposit, with a SRA accession number generated uniquely. Because the data was requested to be deferred to be publicly accessed, at the time of our submission. But now, after double check, we confirm SRP198314 mentioned on Line 138 is a valid accession number, with which all sample information and all sequencing data can be obtained.
Figure 3 the labels are 1-5, but in the text they are referred to as A-E.
Our response: We have now realized the disagreement between labels in figure and that in text can puzzle readers. Label of A-E in Figure 3 represent cluster 1-5. To facilitate the capture of this information, we explained the meaning of label A-E in figure legend of figure 3, as shown on line 201-202.
Finally, there is no confirmation that the DEGs discussed in this manuscript are involved in the shoot differentiation process. Further, there is no compelling story that leads me to believe that these genes are involved in the differentiation process.
Our response: Confirmation of DEGs had been supplemented in Table S2. For all supplementary files needed, we had responded in subsequent questions. For the manuscript, we were focusing on identifying differentiation-related genes, though knowledge for P. euphratica had been mostly summarized from model species, but expression data were learned by using the genetic lines sampled from a natural population. The employment of distinct lines that naturally carrying hugely diverged regeneration capacity, but not using the same lines applying distinct treatment affecting differentiation, such strategy should have great potential to uncover characteristics of shoot regeneration with P. euphratica. This study proposed an initial study on regeneration capacity with P. euphratica, for the further study of regeneration mechanism in depth, our group are launching the quantitative genetic study at population level, hoping the mapping result with natural population can provide us with more precise genes.
Reviewer 2 Report
Some few questions for this manuscript are below:
(1) Can the authors add more detailed information for the four genotypes of HY1, HY2, HY3, and HY4, such as phenotypic description and photo, etc?
(2) Please define A, B, C, D, and E in Figure 3.
(3) A DataSet should be included to list the DEGs. The gene ID of all DEGs mentioned in Figure 5 and Table 1 should appear in the main text.
Author Response
Reviewer #2
Can the authors add more detailed information for the four genotypes of HY1, HY2, HY3, and HY4, such as phenotypic description and photo, etc?
Our response: Yes, phenotypic description of HY1, HY2, HY3, HY4 are the base for the grouping of experiment material. Using phenotypes describing the regeneration capacity, we were able to set the HD and LD group. Their photos had been provided in figure 1. In order to further provide the detailed phenotype, in text we use the regenerated shoot number and number of leaf explant to calculate the mean shoot number per genotype and regeneration rate denoting the regeneration capacity. Please refer to numbers with line 158-164.
Please define A, B, C, D, and E in Figure 3.
Our response: A-E in Figure 3 are subplot label for all expression pattern of all clusters. To avoid the ambiguity of the cluster designation, we had explained what A-E in figure3 denote in legend of figure 3, please see line 201-202.
A DataSet should be included to list the DEGs. The gene ID of all DEGs mentioned in Figure 5 and Table 1 should appear in the main text.
Our response: Thanks for the reminding of supplementary files. We have newly uploaded all supplementary files in submission system of Genes, Table S2 include DEG result comprising gene ID, FDR, log2 fold change, functional annotation, as well as their membership of clustering group, they had contained all genes mentioned in all figures and tables. Also, GO enrichment, KEGG enrichment result were all provided, as provided as Table S3, Table S4 respectively. In revised manuscript, corresponding files were all mentioned, for example, line 198, line 207, line 227.
Reviewer 3 Report
Reviewing the manuscript “Identification of Shoot Differentiation-related Genes in Populus euphratica Oliv.” Submitted to the journal of Genes. In this study, the authors aimed to identify the key regulators in mediating the shoot differentiation in Populus euphratica Oliv. Overall, I do not have any major comment on the manuscript and the study. The manuscript reads well and the necessary controls are carried out. There are few minor comments that should be addressed but aside that I believe the manuscript has the potential for a possible publication.
1- Please include the sequence of the assembled transcripts identified as DEG in supplementary
2- Please provide the Result of the DEG analysis in a supplementary file (Transcript ID, Expression level of the transcript in both conditions, FDR, log2fold change etc.)
3- Please provide the result of the GO and KEGG Enrichment analyses in a supplementary file
4- L104, the link is not working. Please fix the link or provide the genome accession number or the version.
Author Response
Reviewer #3
Please include the sequence of the assembled transcripts identified as DEG in supplementary.
Our response: P. euphratica had been whole genome sequenced, the publicly available resource can be accessed using the gene ID in Table S2. Also, the study takes the strategy of reference-based analysis of transcriptome. Therefore, most reference-based transcriptome research do not include the assembled transcripts with DEGs as supplementary files.
Please provide the Result of the DEG analysis in a supplementary file (Transcript ID, Expression level of the transcript in both conditions, FDR, log2fold change etc.)
Our response: Thanks for the reminding of supplementary files. We have newly uploaded all supplementary files in submission system of Genes, Table S2 include DEG result comprising gene ID, FDR, log2 fold change, functional annotation, as well as their membership of clustering group, they had contained all genes mentioned in all figures and tables. In revised manuscript, corresponding files were all mentioned, for example, line 198.
Please provide the result of the GO and KEGG Enrichment analyses in a supplementary file.
Our response: GO enrichment, KEGG enrichment result were all provided, as provided as Table S3, Table S4 respectively. In revised manuscript, corresponding files were all mentioned, for example, line 207, line 227.
L104, the link is not working. Please fix the link or provide the genome accession number or the version.
Our response: Reviewer #1 might indicate the L140 where a link was found. The ftp link was valid by using ftp software, the normal click cannot simply open the link. To provide convenience for readers, we have corrected this link using an http website of https://www.ncbi.nlm.nih.gov/search/all/?term=Populus%20euphratica, as shown in line 133.
Round 2
Reviewer 1 Report
The authors made every effort to address the concerns I raised in the original submission.